# Chemical Composition and Toxicity of PM_10_ and PM_0.1_ Samples near Open-Pit Mines and Coal Power Stations

**DOI:** 10.3390/life12071047

**Published:** 2022-07-13

**Authors:** Aleksey Larionov, Valentin Volobaev, Anton Zverev, Evgeniya Vdovina, Sebastian Bach, Ekaterina Schetnikova, Timofey Leshukov, Konstantin Legoshchin, Galina Eremeeva

**Affiliations:** 1Department of Genetics and Fundamental Medicine, Institute of Biology, Ecology and Natural Resources, Kemerovo State University, 6 Krasnaya Street, 650000 Kemerovo, Russia; volobaev.vp@gmail.com (V.V.); eugenevdovina@gmail.com (E.V.); sebastianbah430@gmail.com (S.B.); schetnikovakat@gmail.com (E.S.); 2Department of Fundamental and Applied Chemistry, Institute of Fundamental Science, Kemerovo State University, 6 Krasnaya Street, 650000 Kemerovo, Russia; anthonzverev@gmail.com (A.Z.); anchem@kemsu.ru (G.E.); 3Institute of Coal Chemistry and Chemical Materials Science, The Federal Research Center of Coal and Coal Chemistry of SB RAS, 650000 Kemerovo, Russia; 4Department of Geology and Geography, Institute of Biology, Ecology and Natural Resources, Kemerovo State University, 6 Krasnaya Street, 650000 Kemerovo, Russia; tvleshukov@kemsu.ru (T.L.); geology@kemsu.ru (K.L.)

**Keywords:** PM, UFP, air pollution, PM chemical properties, micronucleus, MRC-5

## Abstract

Particulate matter (PM) <10 μm in size represents an extremely heterogeneous and variable group of objects that can penetrate the human respiratory tract. The present study aimed to isolate samples of coarse and ultrafine PM at some distance from polluting industries (1–1.5 km from the border of open-cast mines). PM was collected from snow samples which allowed the accumulation of a relatively large amount of ultrafine particles (UFPs) (50–60 mg) from five objects: three open-cast mines, coal power plants, and control territories. The chemical composition of PM was examined using absorption spectroscopy, luminescence spectroscopy, high-performance liquid chromatography, X-ray diffraction (XRD), and X-ray fluorescence (XRF) analyses of solid particle material samples. Toxicity was assessed in human MRC-5 lung fibroblasts after 6 h of in vitro exposure to PM samples. The absorption spectra of all the samples contained a wide non-elementary absorption band with a maximum of 270 nm. This band is usually associated with the absorption of dissolved organic matter (DOM). The X-ray fluorescence spectra of all the studied samples showed intense lines of calcium and potassium and less intense lines of silicon, sulfur, chlorine, and titanium. The proliferation of MRC-5 cells that were exposed to PM_0.1_ samples was significantly (*p* < 0.01) lower than that of MRC-5 cells exposed to PM_10_ at the same concentration, except for PM samples obtained from the control point. PM_0.1_ samples—even those that were collected from control territories—showed increased genotoxicity (micronucleus, ‰) compared to PM_10_. The study findings suggest that UFPs deserve special attention as a biological agent, distinct from larger PMs.

## 1. Introduction

Mining activity is inevitably followed by aerosol emissions, which include particulate matter (PM). During mining, most PM is generated owing to the mechanical disintegration of coal and maternal rock. Moreover, PM is also generated during explosion and combustion processes. Emissions also include gaseous components and droplets that can accompany PM in potential exposure during inhalation and airway exposure. This means that PM is a multipollutant [1]. Adverse effects can be synergistically amplified, but the model of cell–pollutant interactions is unclear.

The features of the surrounding natural and anthropogenic landscape can affect the amount of PM generated at any point. An increase in humidity can reduce PM emissions, whereas technical processes, such as drilling and explosion processes, topsoil removal, coal hauling, and stockpile erosion, can increase PM emissions [2]. Fine particles with a diameter of 0.8–2 microns, which are abundant in metals in poorly soluble forms, comprise the second most predominant component of the products formed during the incomplete combustion of coal and are possibly contained in fuel oil. In contrast, ultrafine particles <0.1 μm (UFP) are formed during the complete combustion of fuel, and such particles mainly contain water-soluble metal sulfates [3]. The spread of dust can vary from season to season. In a study area, a decrease in temperature and average wind speed in winter leads to a reduction in the mixing of air masses and PM spread in the lower atmospheric boundary layers. In summer, high wind speeds promote the mixing of air layers and PM spread. The concentration of PM in the vicinity of the object in the winter exceeds the summer concentration. At the same time, a previous study of coal mines located in northern India reported that the concentration of the UFP does not strongly depend on weather conditions and that even prolonged monsoon rains do not lead to a rapid decrease in PM_1.0_ in air [4].

In regions that are in close proximity to coal mines, the geological features and properties of coal can considerably affect the amount and quality of generated dust [5]. The main characteristics of the coal and surrounding rocks are their brittleness and hardness. An increase in brittleness leads to an increase in the proportion of fine and respirable PM and decreases the mass fraction of dust relative to the mass of coal and rock [5]. Coal-bearing rocks are characterized by a high level of erosion and the ability to form respirable PM_10_ and PM_2.5_. In addition, and to a greater extent, this applies to lignite, which increases dust formation several times owing to wind erosion as well as during transportation [6]. Moreover, the incomplete combustion of coal leads to the generation of coarse PM_10_, which includes coal particles and inorganic calcium aluminosilicate fragments.

Coal storage facilities and dumps are also an important source of PM, and rock particles with lower moisture levels can generate more dust than coal itself [7]. The transportation of coal by rail and road remarkably increases the local load of PM, especially that of respirable PM_3_ and PM_10_. Even if the distance of such transfer events is not great (50–100 m), the roads can be located close to residential houses and offices, and the generated PM can directly affect people [8]. When coal and rocks are transported using heavy equipment, PM_10_ is generated in an amount comparable to that with the generation of PM near a mine [9].

Research in northern Colombia has shown that in a region with a high density of open-pit coal mines, the “share of the impact” (i.e., some dust particles with aerodynamic sizes <10 μm), relative to the initial volume of dust emissions (resulting in lung exposure among residents), was 6 × 10^−9^ to 4 × 10^−8^ [10]. The authors acknowledged that the obtained values were high in comparison with those in a number of similar studies. However, the generation of PM at the studied sites can be quite high, given the high intensity of production (5–9 million tons of coal per year at each open-pit mine).

Aerodynamic parameters determine the range of transmission of PM. Coarse particles from 2.5 to 10 μm (PM_10_) are deposited in the upper respiratory tract, whereas the fine fraction, with sizes of 0.1–2.5 μm (PM_2.5_) penetrates the lower respiratory tract and alveoli [11]. The PM_2.5_ fraction is considered the most dangerous and is capable of causing chemical reactions in the alveolar areas and the transport of toxic components to the alveolar wall [12,13]. Technological changes have led to the proliferation of new anthropogenically clean coal technologies that limit fine particles and nitrogen oxide emissions, which in turn causes a sharp increase in the proportion of UFPs in the total PM emissions. Furthermore, fossil fuel power plants, refineries, and smelters constitute the leading sources of UFPs in the lower atmosphere at the surface [14]. Their small size allows UFPs to travel through the respiratory system to the alveoli and even through the alveolar–capillary barrier. The surface area of UFPs can be up to 100 times higher than that of coarse particles of comparable mass; moreover, UFPs can have a high potential for the absorption of heavy metals and other accompanying pollutants compared to that of coarse particles [15,16].

The distribution of nanoparticles and their impact on human health have not been sufficiently studied in comparison with those of larger particles. In many reviews similar to the study conducted by Kwon et al. [15], the main problems associated with PM exposure were investigated. These reviews have focused on a certain question, for example, which indicators best characterize PM exposure, specifically the number of particles or their mass. Whether the effects of condensed and solid PM differ, the toxicity of UFPs smaller than 10 nm comprises the delayed effects of PM [15]. PM less than 10 μm in size, which is capable of reaching the human respiratory tract, represents an extremely heterogeneous and variable group of objects. Large particles affect the external respiratory tract and bronchi, whereas UFPs penetrate the lower respiratory tract and can even enter the systematic circulation system. Hence, UFPs deserve attention as biological agents, distinct from large PMs. Particles less than 100 nm in size are characterized by a high particle count and large surface area. This provides a high capacity for the adsorption of organic or inorganic pollutants, including reactive oxygen species. When evaluating the properties, it is necessary to consider both the mass of the particles and their surface area. The PM characteristics can differ significantly in PMs of different origins, following a variation in their fractional composition.

UFP exposure is often discussed in connection to cellular inflammatory responses and oxidative stress induction, and the pathological responses of lung tissue can include epithelial–mesenchymal transition [17]. UFP exposure can change gene expression. Diesel- and biomass-combustion-generated UFPs can induce the expression of inflammatory markers and transcriptional markers relevant to cardiovascular disease [18], alter the mitochondrial metabolism, and induce cytochromic genes [19] in human BEAS-2b cells. Furthermore, ultrafine carbonaceous particles can induce heme oxygenase-1 but not IL-6 and IL-8 [20], and diesel exhaust UFPs can increase metal oxide cytotoxicity [21] in A549 human lung adenocarcinoma cells. Moreover, reactive oxygen species’ generation following UFP exposure was demonstrated in mouse pulmonary microvascular endothelial cells [22], and protein oxidation and DNA adducts were identified in mice in vivo [23]. The genotoxic and other adverse effects of dust pollution are being actively studied. Exposure to PM generated by coal mines and coal power plants has been reported to increase micronucleus and chromosomal aberrations’ frequency [24,25,26,27,28,29,30].

It is important to investigate the particles of inhalable PM which are transferred a considerable distance from coal mines and to which not only professional but also neighboring groups of people can be exposed. Metals, polycyclic aromatic hydrocarbons (PAHs), and quartz particles can also be found in different proportions in PM samples [31]. The actual black carbon (BC) particles are usually assumed to be separate from PM. BC can be larger than 10 µm and is often considered as more than 100 nm particles [32,33]. The present study aimed to isolate samples of coarse and ultrafine PM at a certain distance from coal mines (1–1.5 km from the border of an open-cast mine). The distance was determined by the state standard (500 m) and residential development, for which there is a high density of buildings at this distance from the coal industry. We hypothesized that these samples would reflect the exposure based on spread along the coal-hauling roadways via air currents from the object exposing residents of nearby settlements. Using the snow sample method, we successfully collected a relatively large amount of UFPs (50–60 mg). This high concentration facilitated a simulation of conditions of PM exposure to cells in vitro. It also helped to neutralize the effect of fluctuations in the concentration of dust on different days.

## 2. Materials and Methods

### 2.1. Collection and Extraction of PM

Particle samples were collected from the territory of the Kemerovo region (western Siberia, Russia) at the end of the snow accumulation period (March 2020). Samples were collected at a distance of 1–1.5 km from the coal industry facilities located outside the sanitary protection zone. This distance was selected based on the dominant direction in which the wind rose in winter (determined using Google Maps on the pollution of the snow). The sampling points were situated near the open-pit mine “Bachatskiy,” the coal mines “Novobachatskiy” and “Permyakovsky,” and the coal-fired power plant “Belovskaya” in the town of Belovo (Russia) (Figure 1). PM samples that were used as controls were collected near the village of Krasnoe, which is situated at a considerable distance from the industrial enterprises. All of the surveyed enterprises are located in the vicinity of residential settlements, which can directly affect the quality of atmospheric air, including in relation to particulate matter. These mines produce a total of about 15 million tons of coal (“Permyakovsky”—approximately 5.2; “Novobachatskiy”—design capacity of about 0.5 million tons; “Bachatskiy”—approximately 9.5 million tons.). The coal-fired power plant uses about 2.8 million tons of coal per year. The coordinates and symbols of the PM collection points are listed in Table 1.

Sampling was carried out using the snow survey method (GOST 17.1.5.05-85) on a plot of 5 × 5 m. With this method, snow samples are collected using snow sampling devices, which are made of chemically resistant polymer material. Debris is removed, and soil particles are prevented from entering the samples.

The collected samples were placed in clean, sealed containers made of a chemically resistant polymer material. Ten snow samples were collected from each sampling area; these samples were first thawed and then mixed in equal proportions in the laboratory to obtain a weighted average contamination component. After thorough mixing, 1 L of liquid was taken from the collected samples and frozen.

The frozen samples were thawed and subjected to sequential filtration using a Sterifil vacuum system (Merck KGaA, Darmstadt, Germany) on membrane nylon filters with different pore diameters (10, 2.5, and 0.1 μm) (GVS, Sanford, FL, USA). The first filtration (10 µm filter) removed particles >10 µm. Using the second filtration (2.5 µm filter), PM_10_ particles (2.5–10 µm) were collected from the filters.

The third filtration (0.1 µm filter) process was used to collect PM_2.5_ particles (0.1–2.5 µm). The liquid that passed through all filters was the basis for obtaining a suspension of nanofraction PM_0.1_ (particles less than 100 nm). Suspensions of PM_10_ and PM_2.5_ fractions were obtained by removing the fractions from filters in an ultrapure aqueous medium (Simplicity, Merck KGaA, Darmstadt, Germany) using an Elmasonic S30H ultrasonic bath (Elma, Singen, Germany).

The suspensions were partially concentrated using an Eppendorf Concentrator plus a vacuum rotary concentrator (Eppendorf, Hamburg, Germany), transferred into pre-weighed microtubes, and concentrated until they were dry. The tubes were weighed again on an Ohaus Pioneer PX125D semi-micro balance with an ION-100A ionizer (Ohaus, Parsippany, NJ, USA). The weight of the dry residue of the fraction was determined from the difference in the weight of the test tube before adding the suspension and the weight after adding the suspension. The concentrated suspension was dissolved in Hanks’ solution (Biolot, Saint Petersburg, Russia) to obtain a particle concentration of 10 mg/mL. Thereafter, it was disinfected using ultrasound in an ultrasonic bath for 30 min.

A suspension of ALEX nanosized aluminum powder (Advanced Powder Technologies, Tomsk, Russia) was used as a positive control. The aluminum powder sample was sterilized at 121 °C for 30 min in a standard mode. Thereafter, it was vortexed and treated with an Elmasonic S30H ultrasonic bath (Elma, Singen, Germany) for 5 min. This suspension was used as a positive control for the cell exposure.

### 2.2. Physical and Chemical Analysis of the Melting Snow and Particulate Matter Samples

The absorption spectroscopies of the melted snow samples which were concentrated eight times were registered using a scanning spectrophotometer, Shimadzu UV-3600 (Shimadzu Corporation, Kyoto, Japan). Spectra were recorded using a 1 cm quartz cuvette for a wavelength range of 190–1300 nm with 0.2 nm step.

Luminescence spectra of the melted snow samples were measured using a scanning spectrofluorometer (“Fluorat-02-Panorama”, Lumex, Saint Petersburg, Russia) in a 1 cm quartz cuvette.

For high-performance liquid chromatography, a 25 mL aliquot of the sample and 5 mL of hexane were added to a flask, and extraction was carried out twice using a magnetic stirrer for 30 min. Thereafter, the contents of the flask were allowed to settle for 10 min to separate the solvent layers completely. The hexane extract layer was transferred into a 50 cm^3^ conical flask which was fitted with a ground-in stopper. The collected extract was stored for 2 h in a freezer at minus 12–24 °C. This procedure ensured the conversion of water remaining in the extract into ice, which was formed on the flask walls, thereby allowing the separation of residual water from hexane. After the separation, the obtained hexane extract was quickly poured into a clean flask, and it was evaporated in a sand bath at 60 °C to obtain a residual volume of 0.5–1 mL. The residue was transferred into a 5 mL volumetric tube and evaporated at room temperature until traces of hexane were barely visible. At the end of the evaporation, 0.5 mL of acetonitrile was added along the walls of the test tube. The resulting solution was transferred to a chromatography vial. The samples were then analyzed twice on a Shimadzu LC-20 high-performance liquid chromatograph with acetonitrile/water (50/50) eluent for 30 min on a PerfectBond ODS-HD C-18 column (250–3.0 mm (5 μm)) using a spectrophotometric diode-matrix detector (the elution mode was gradient, and the flow rate was 0.5 mL/min). Raman spectra of the particulate matter samples were measured using the Raman Spectrometer LabRAM HR (Horiba Scientific, Piscataway, NJ, USA). Laser radiation with wavelengths of 633 and 784 nm was used for excitation. 

### 2.3. XRD and XRF Analysis of Solid Particle Material Samples

The DRON-8 (BOUREVESTNIK JSC, Saint Petersburg, Russia) and Difray-401k (JSC Scientific Instruments, Saint Petersburg, Russia) diffractometers were used to determine the phase composition of the PM samples. The XRF module measured the X-ray fluorescence spectra for Difray-401k based on an Amptek X-123SDD X-ray spectrometer (Amptek Inc., Bedford, MA, USA).

### 2.4. Cell Cultivation and PM Exposure

In the present study, a diploid cell culture of human embryonic lung fibroblasts MRC-5 (passage 20) (ATCC, Manassas, VA, USA) was used as the model. It was obtained from the collection of the State Scientific Center of Virology and Biotechnology “Vector” (Novosibirsk, Russia) in frozen form. All cell culture manipulations were performed in a clean cell box in a BAVnp-01 laminar flow cabinet (LAMSYSTEMS, Miass, Russia). Cultivation was carried out in a 5% CO_2_ incubator, Binder CB 53 (Binder, Tuttlingen, Germany).

The preliminary growth of the cell mass (23–24 passages) was carried out in 75 cm^2^ culture flasks (Eppendorf, Hamburg, Germany). Cell cultures were grown on Igla-MEM nutrient medium (89%) using a solution of non-basic amino acids (1%) and fetal calf serum (10%) (Biolot, Saint Petersburg, Russia). Penicillin–streptomycin (Paneco, Moscow, Russia) was used as an antibiotic at a concentration of 25,000 U per 500 mL of medium. The inoculum concentration following the recommendations of the cell culture supplier was 175 thousand cells per 1 mL of medium. The cultivation period was 72 h.

To remove cells from the monolayer, the culture medium was removed, washed with trypsin–versene solution (1:1) (Biolot, Saint Petersburg, Russia), and then incubated with a new portion of trypsin–versene for 5 min. The reaction was stopped by adding fresh nutrient medium, after which, the cell suspension was transferred into sterile tubes. Pellets of cells were created using centrifugation (5 min at 150× *g*), and the supernatant was removed. Fresh medium (1 mL of fresh medium) was added to the cell pellet, followed by pipetting and extracting 20 μL of the sample to count the number of cells.

The number of cells was stained with Trypan blue dye (Biolot, Saint Petersburg, Russia) to identify non-viable cells, and thereafter, the cells were counted using a Goryaev camera. The counting of the number of cells per 1 mL was carried out according to the formula 250,000 × the average number of cells in the large square of the Goryaev chamber. The semi-lethal and semi-cytostatic PM concentrations were determined previously. For this, the culture was planted in 6-well (9 cm^2^ well area) culture plates (Eppendorf, Hamburg, Germany), after which, it was cultured for 24 h.

The final concentration of PM samples included 0.25, 0.5, 1, and 2 mg/mL, which were cultured for 6 h. Thereafter, the growth medium was removed, the monolayer was washed with fresh medium to remove PM, and fresh medium was added and cultured for another 48 h. During the gradient experiment, three replicates were cultured for each sample type. After determining the semi-lethal and semi-cytotoxic concentrations, the optimal concentration of the experimental load was determined to be 0.5 mg/mL.

The cell culture was also planted in six-well (with well areas of 9 cm^2^) culture plates (Eppendorf, Hamburg, Germany), after which, the cells were cultured for 24 h. The cells were cultured for 6 h, and the growth medium was removed. The monolayer was washed with fresh medium to remove PM, fresh medium was added, and the cells were cultured for another 48 h. At the same time, samples of negative (“negative control”), positive (“positive control”), and dilution control (“dilution control”) were set.

The volumes of the culture medium equivalent to those of the PM samples were added to the negative control samples. AlEX powder was added to the positive control samples at equivalent PM concentrations. Hanks’ solution in a volume equivalent to that of the PM samples was added to the dilution control samples. During one experiment, six wells were cultured for each type of sample. Each experiment was independently repeated three times. 

### 2.5. Viability Assessment and Micronucleus Test

Cell survival was calculated as the proportion (%) of viable cells to the total number of cells determined in the sample using the Goryaev chamber. The “Relative Increase assessed cytotoxicity in Cell Count” (RICC) [34] was calculated using the following equation:(1)$$\mathrm{RICC}\;=\frac{\;\mathrm{Increase}\;\mathrm{in}\;\mathrm{the}\;\mathrm{number}\;\mathrm{of}\;\mathrm{cells}\;\mathrm{in}\;\mathrm{the}\;\mathrm{exposed}\;\mathrm{culture}\;(\mathrm{cell}\;\mathrm{number}\;\mathrm{in}\;\mathrm{the}\;\mathrm{end}\;-\;\mathrm{cell}\;\mathrm{number}\;\mathrm{at}\;\mathrm{the}\;\mathrm{start})\;}{\;\mathrm{Increase}\;\mathrm{in}\;\mathrm{the}\;\mathrm{number}\;\mathrm{of}\;\mathrm{cells}\;\mathrm{in}\;\mathrm{negative}\;\mathrm{control}\;\mathrm{samples}\;(\mathrm{cell}\;\mathrm{number}\;\mathrm{in}\;\mathrm{the}\;\mathrm{end}\;-\;\mathrm{cell}\;\mathrm{number}\;\mathrm{at}\;\mathrm{the}\;\mathrm{start})}\times 100$$

For the micronucleus test (MN), 10^5^ cells obtained after removing the experimental samples were used. Cell suspensions were fixed using cold Clark’s fixative (methanol:acetic acid, 3:1). The fixed suspension was mounted onto cold, defatted glass slides and dried.

The preparations were stained with a fluorescent dye 4′, 6-diamidino-2-phenylindole (DAPI, 0.5 μg/mL, pH 5.0) using a Fluroshield mounting medium (ABCAM, Cambridge, UK). The staining procedure was performed as follows: one small drop (approximately 15 μL) of Fluroshield was placed on the slides, which were then covered with cover slips.

Thereafter, the preparations were evaluated using an Altami LUM 1 fluorescent microscope (Altami, Saint Petersburg, Russia) at a magnification of × 1000. The glass slides were encrypted to exclude the subjectivity of microscopy. Cells with micronuclei were counted regardless of the number of micronuclei (“MN”). Moreover, 2000 cells were analyzed for each slide to record the states of the “Nuclear bridge” type (“NB” indicator) and “Nuclear protrusion” (“NP” indicator).

### 2.6. Statistical Analysis

Statistical analyses were performed using the Statistica 10.0 software package (StatSoft, Tulsa, OK, USA). The mean values and standard deviations were calculated for each parameter. Compliance with the normal distribution was determined using the Kolmogorov–Smirnov test. The presence of differences in the levels of the studied parameters between samples treated with particles from different points was determined using the Kruskal–Wallis rank test. The comparison of parameters between samples “nano” and “micro”, as well as pairwise comparison of the parameters of samples from different points, was carried out using the Mann–Whitney test.

## 3. Results

### 3.1. Physicochemistry Properties of the Melting Snow and Particulate Matter Samples

Table 2 presents the absorption spectra of the samples of melted snow used in the present study. The spectra of P, N, K, and C samples recorded bands that showed maxima at 200 nm, corresponding to the absorption of nitrate ions in water. No nitrate ion band was observed in sample B (all samples are enlisted in Table 1). The highest nitrate ion content was observed in sample C.

All samples recorded a wide non-elementary absorption band with maxima at 270 nm. Upon excitation at a wavelength of 270 nm, a broad luminescence band with a maximum close to 450 nm was observed in the luminescence spectra of all the samples (Table 2). This band can also be associated with dissolved humic compounds [35]. A second intense luminescence band with a maximum at approximately 325 nm was observed in the spectrum of sample B. This band was absent in the luminescence spectra of all other samples, and it can be associated with the luminescence of contaminating components, for example, oil products dispersed in solution or adsorbed on the surface of dispersed particles [36,37].

Intense lines of Ca and K and less intense lines of Si, S, Cl, and Ti were observed in the X-ray fluorescence spectra of all the studied samples. The spectra of micron powders B_10_, K_10_, and P_10_ had the most intense Ti lines. In the nanosized samples, N_0.1_ and B_0.1_, clear lines of Sr and Al were observed, whereas in C_0.1_, only a weak line of Sr was observed. However, no Si line was observed in any of these samples. Cr bands were observed in the micron-sized samples P_0.1_, B_0.1_, K_0.1_, and C_0.1_. Sample B_10_ also contained bands of Zn, Mn, and Pt, whereas only Mn was observed in a fraction less than 100 nm. Zn bands were also observed in sample C. The content of Fe in the samples could not be determined from the obtained spectra because a lamp with an iron anode was used for excitation in the present study.

A considerable component of all the studied samples included the amorphous phase, as evidenced by the results of X-ray phase analysis and Raman spectroscopy (Table 2). All the micron-sized samples contained quartz. Calcite reflections were detected in the diffractograms of micron-sized samples B_10_, N_10_, and P_10_ and in the diffractograms of all nanosized samples except C_0.1_. The only crystalline constituent of C_0.1_ was gypsum. Crystalline phases of sodium and potassium chlorides were also observed in N, K, and P samples. In the Raman spectra of all microfractions except C_10_, the D and G modes of amorphous carbon were observed, which may indicate the presence of coal particles or products of the combustion of carbon energy carriers. In the nanofraction, these bands were absent, which indicates that all carbon particles present in the samples were >100 nm.

Crystalline particles present in the samples, except C, were represented by calcite polycrystalline quartz and amorphous silicate phases, which have a developed surface. Quartz and amorphous carbon—which had particles bigger than 100 nm—disappeared in PM_0.1_ samples compared to PM_10_. PM_10_ can also adsorb humic compounds, polyaromatic components, and other toxic components of the samples. Thus, the genotoxic properties of particles can be attributed to either the components of the solid part of the particle itself or by substances adsorbed on its surface or both.

The positive control included a mixture of Bayerite (PDF 20-0011) Al(OH)_3_(Al_2_O_3_·3H_2_O) and boehmite (PDF 5-0190) Al_2_O_3_·H_2_O. The incompletely dehydrated aluminum hydroxide boehmite reflections were wide at half height. The crystallite size of approximately 2–3 nm was estimated for the positive control using the Scherrer equation.

### 3.2. High-Performance Liquid Chromatography

The following polyaromatic hydrocarbons were identified in almost all water samples obtained from the snow cover: anthracene, benz[a]anthracene, benz[a]pyrene, benz[b]fluoranthene, benz[k]fluoranthene, biphenyl, dibenz[a, h]anthracene, pyrene, phenanthrene, fluoranthene, fluorene, chrysene, and 2-methylnaphthalene (Table 3).

The highest total concentration of PAHs was found in sample B. PAH concentrations in samples C and P were equal to and 2.5 lower than the PAH concentration in sample B, respectively. Sample N had the lowest concentration, which was lower by orders of magnitude compared to that of sample B. The dibenzo[a, h]anthracene (Group 2A “Probably carcinogenic to humans” by IARC), pyrene (Group 3 by IARC), benzo[a]anthracene (Group 2B “Possibly carcinogenic to humans” by IARC) levels were higher in sample B, and sample B also contained a high concentration of phenanthrene (Group 3 by IARC).

### 3.3. Viability Assessment and Micronucleus Test

At a PM sample final concentration of 1 mg/mL, the survival rate of cells in all samples was >70%, but the RICC for some samples was <40 (Table 4).

It was also noted that at 1 mg/mL PM dosages, the RICC values in the samples loaded with nanoparticles were significantly lower than those in the samples loaded with microfractions (*p* < 0.01). To further study the genotoxic properties of PM, the cells were cultured with a PM concentration of 0.5 mg/mL (Table 5).

A significant increase in the frequency of genotoxic effects in the positive control samples was observed compared with the negative control samples (*p* ≤ 0.001). Moreover, no significant difference was observed between the negative control and dilution control. Thus, it can be concluded that the cell model was accurate.

On comparing negative control samples with samples exposed to PM, significant differences in the studied parameters were noted between the negative control and the nanofraction “C” (*p* = 0.002) and “B” (*p* ≤ 0.008), in the absence of differences with a microfraction with the same points. All parameters of samples N, K, and P differed from those of the negative control samples for both nano- and microfractions.

On comparing the studied parameters between the samples exposed to PM_10_ samples from different points, a significant difference in the parameters “MN” (*p* = 0.0017 H = 17.24), “NP” (*p* < 0.0001 H = 28.78), and “NB” (*p* = 0.0002 H = 22.54) was observed. When comparing the parameters between the samples exposed to PM nanofractions from different points, a significant difference in the “MN” parameter (*p* = 0.0005 H = 19.97) was observed.

In PM_10_ that was sampled from collection point “C” (control without industry), the exposure samples had significant differences compared with points “B” (*p* ≤ 0.02), “N” (*p* < 0.0001), and “K” (*p* ≤ 0.002), by MN, NP, and NB and with “*p*” points (*p* < 0.001) for MN and NB.

Significant differences were observed between PM_0.1_ samples collected from the control point “C” and those collected from point “B” (MN and NB, *p* = 0.02), point “N” (NP, *p* = 0.028; NB, *p* = 0.004), point “K” (NB, *p* = 0.047), and point “*p*” (NP, *p* = 0.01). However, not all the differences in PM_0.1_ remained significant after false discovery rate (FDR) correction. In contrast, all the differences calculated for PM_10_ remained significant after FDR correction.

On comparing the parameters of the genotoxic effect in PM_0.1_ and PM_10_ samples from one point, we observed a significant difference (passed the FDR correction) between the levels of genotoxic effects for the samples collected from points “C” (all parameters *p* < 0.002), “B” (MN and NB, *p* < 0.002), and “P” (MN and NP, *p* < 0.004).

## 4. Discussion

All samples recorded a wide non-elementary absorption band with maxima at 270 nm. Such absorption bands are usually associated with the absorption of dissolved organic matter (DOM) [35], which is primarily represented by humic compounds. However, this spectral region may also represent the absorption of contaminants, such as polyaromatics. This absorption band had the lowest intensity for samples C and B, whereas it had the highest intensity for sample N. A downturning wing extending to the long-wavelength region, which was most noticeable in samples B and N, can be related to the scattering and absorption of solid particles, including colloidal matter. The short-wavelength part of this band can also be associated with the absorption of dissolved compounds. The source of nitrate in all samples can be soil particles brought from agricultural fields and containing nitrate fertilizers adsorbed on them. Usually, the most abundant elements in PM composition are Fe-, Al-, Cu-, and Si-rich particles [38]. PM contains mineral matter, crystalline particles, elemental carbon, and organic matter [13], as well as an enrichment of multiple elements in respirable dust PM_0.1_ [39].

UFPs are considered highly toxic substances with higher concentrations of volatile and absorbed compounds, with enrichments up to 50-fold in UFPs against coarse or fine fractions [40]. The cited authors also mentioned a possible increase in the number of oxygenated functional groups on the surface, which correlated with the high carbonaceous content of the UFPs.

Additionally, in open-pit mines, extracted coal is stored in sunlight at high ambient temperatures, where spontaneous and incomplete coal combustion may lead to PAH emissions [41,42]. Particularly in open–cast mining facilities, these toxic substances are released into the atmosphere, where they can form complex mixtures (mineral fraction/black coal/vehicle exhaust/explosive components) [43]. The components of such mixtures can have potentially synergistic effects, and hence, such mixtures represent an essential health and safety hazard to exposed populations [44,45].

Despite several disadvantages (different times that particles stay in snow or long periods of preliminary concentration to obtain sufficient amounts of PM samples), the snow survey method makes it possible to minimize fluctuations in the concentration of particles in the air caused by industrial and climatic factors. It can be assumed that the composition of PM collected on the filters in this method notably differs based on the time and day. On the one hand, this creates difficulties in interpreting the results and comparing them with similar studies. On the other hand, the accumulation of PM and its possible chemical modification in snow cover may be followed by the repeated transfer of fine and ultrafine particles and possible exposure. This PM modification aspect seems difficult to assume but essential for understanding the problem in the long-term accumulation/erosion of PM.

The high total surface area of PM particles increases their ability to absorb organic substances and metal ions. The presence of several organic compounds in the aromatic series, which could be adsorbed on the PM, were observed in the PM samples that were collected in the present study. At the same time, the absorption spectra obtained for PM samples diluted with deionized water showed the presence of a significant amount of organic materials.

The release of a large volume of PM can lead to a noticeable decrease in the quality of life and welfare and lead to substantial losses, expressed in both medical and economic indicators [46]. Vehicle traffic on access roads has been identified as the largest source of flying dust and may even contribute 80% of the total dust emissions [47]. Moreover, rugged terrain in a region often motivates residents to build their homes close to the roads of coal mines [9].

We assumed that a sorption mechanism of substances on the PM is present, which facilitates qualitative changes in chemical composition and subsequent biological effects. Natural objects of PM, which expose humans and animals, are likely to one way or another undergo a similar period of sorption; before entering the respiratory tract, they are in the upper layers of soil, snow, or leaf surfaces. The material for modification near industrial facilities can be crushed coal seams, photochemical processes on the surface of particles, blasting products and other mining operations, and vehicle engine exhaust. The joints on the PM surfaces were also modified using UV radiation. Such a scenario seems quite likely within a few kilometers from the source of the PM. Thus, snow-sampling-based research can be useful to verify this suggestion. As snow is a combination of a crystalline phase and a liquid phase of a water film on the surface of ice crystals, a similar scenario can be probably realized in regions without a long periods of snow cover.

A review of these data, among other sources of pollution, is presented by da Silva [48]. An increase in cytokinesis-block micronucleus CBMN frequency is expected to positively correlate with work experience under exposure to coal dust [25,26]. An increase in the level of DNA damage is typical for a situation of mixed pollution, including coal dust, bottom ash, and fly ash [49], and has been confirmed via studies of natural objects. The micronucleus and comet assays of samples of Mus musculus and Iguana iguana collected from coal mining fields revealed a high level of DNA damage compared to the samples obtained from sites that were far from the source of coal dust [50].

The exposure of residential dust particles to open-pit coal mines also shows some indicators of DNA damage. This includes an increase in % Tail DNA level in the alkaline and the modified Comet assay and DNA fragmentation in buccal cells’ cytome assays, cytokinesis (binucleated cells), and cell death (condensed chromatin, karyorrhexis, pyknosis, and karyolysis) [45]. In addition, an increase in micronucleus (MN) frequency was observed in binucleated (MNBN) and mononucleated (MNMONO) cells, and a positive correlation between centromere-positive MN and PM_2.5_ levels in residents’ proximity to coal mining activities was found [41].

## 5. Conclusions

The objective of this study was to isolate PM less than 10 µm in size, isolate the PM_0.1_ fraction of nanoparticles separately, describe its physicochemical properties in detail, and expose these particles to a biological model in vitro. The use of snow samples made it possible to reduce fluctuations in the composition of particles and to isolate a large amount suitable for the experiment. The isolated particles were predominantly crystalline with a significant content of DOM; PAH was not found in particles but was found in the dissolved phase, possibly migrating from the surface. Carbonaceous particles were not found among nanoparticles, and amorphous carbon was not able to form particles smaller than 100 nm according to our data. An evaluation of the biological effects showed an increase in the frequency of micronuclei and a decrease in proliferative parameters in samples exposed to PM_0.1_ relative to those in samples exposed to PM_10_. A slight increase in micronuclei was found in samples B and N, containing the mineral fraction of quartz and calcite and a number of metals. This, however, did not significantly distinguish them from other samples, except for the high content of PAH in sample B. We suggest that UFPs are distinct from larger PMs, as they comprise a biological agent that does not contain particles of amorphous carbon, and that the source of toxicity is the mineral phase.

## Figures and Tables

**Figure 1 life-12-01047-f001:**
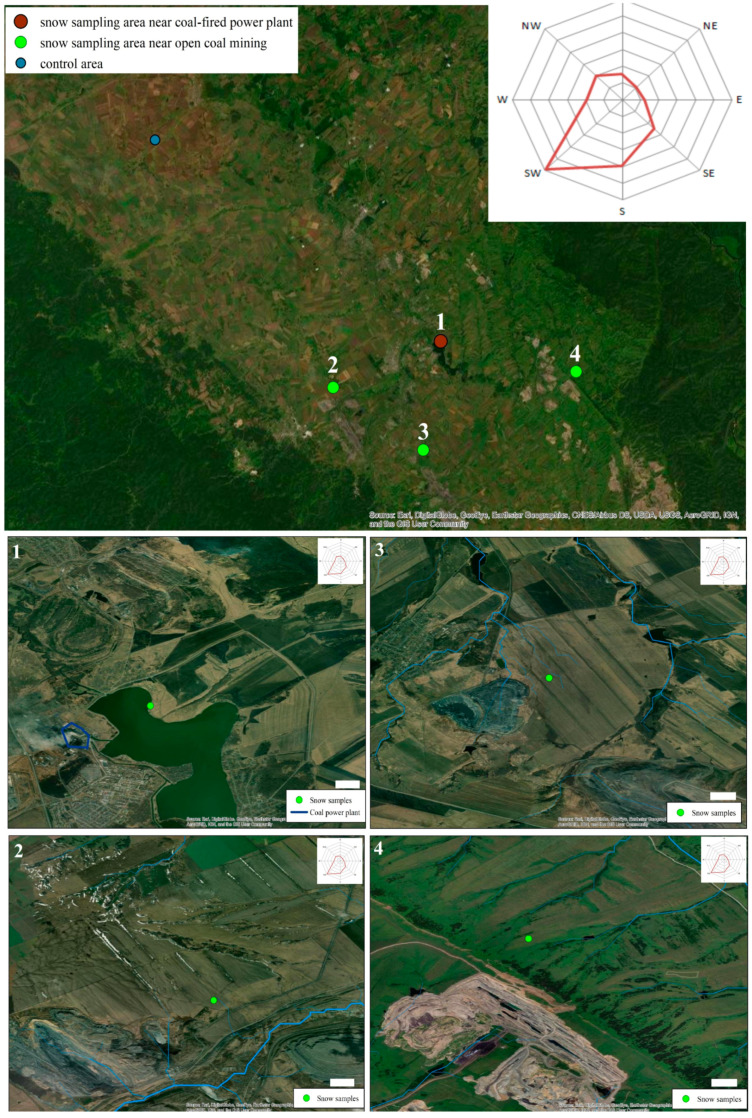
Location of sample collecting points.

**Table 1 life-12-01047-t001:** The coordinates and symbols of PM collection points.

PM Collecting Point	Coordinates	Symbol
Near v. Krasnoe	N 54°49.976′ E 85°30.363′	C
Open-pit mine “Bachatskiy”	N 54°21.454′ E 86°07.030′	B
Open-pit mine “Novobachatskiy”	N 54°14.774′ E 86°23.112′	N
Coal-fired power plant “Belovskaya”	N 54°27.165′ E 86°26.880′	P
Open-pit mine “Permyakovsky”	N 54°23.802′ E 86°53.972′	K

**Table 2 life-12-01047-t002:** Phase and elemental composition of suspended particles of different sizes.

		Sample				
		C	B	N	K	P
XRD, Raman spectroscopy	PM_10_	Quartz (SiO_2_) (very weak)	Quartz (SiO_2_) Calcite (CaCO_3_) Carbon	NaCl Quartz (SiO_2_) Calcite (CaCO_3_) Carbon	Quartz (SiO_2_) NaCl Carbon	Quartz (SiO_2_) Calcite (CaCO_3_) Carbon
	PM_0.1_	Gypsum CaSO_4_·2H₂O	Calcite (CaCO_3_)	Calcite (CaCO_3_) NaCl, KCl	Calcite (CaCO_3_)NaCl	Calcite (CaCO_3_) KCl
XRF	Joint	K, Ca, Si, S, Cl, Ti				
	PM_10_	Cr	Ti, Cr, Cu, Zn, Mn, Pt	Cu (weak)	Ti, Cr	Ti, Cr, Cu (weak)
	PM_0.1_	Sr (weak), Cu, Zn, Hf	Al, Sr, Mn	Al, Sr, Cu (weak)	Zr	Cu (weak), Rn

Note: XRD—X-ray diffraction, XRF—X-ray fluorescence.

**Table 3 life-12-01047-t003:** PAH content in analyzed water samples with an accuracy index, δ, % (*p* = 0.95).

No	PAH	B		P		C		N		K	
		C, mkg/L	±δ, %	C, mkg/L	±δ, %	C, mkg/L	±δ, %	C, mkg/L	±δ, %	C, mkg/L	±δ, %
1	Anthracene	0.16 ± 0.07	43	0.14 ± 0.06	43	0.12 ± 0.05	43	0.32 ± 0.14	43	0.14 ± 0.06	43
2	Benz[a]anthracene	9.40 ± 2.73	29	5.50 ± 1.60	29	3.80 ± 1.67	44	1.20 ± 0.53	44	-	-
3	Benz[a]pyrene	0.36 ± 0.15	43	0.49 ± 0.21	43	0.23 ± 0.10	43	-	-	-	-
4	Benz[b]fluoranthene	5.10 ± 1.53	30	2.65 ± 1.17	44	1.90 ± 0.84	44	0.22 ± 0.10	44	-	-
5	Benz[k]fluoranthene	3.65 ± 0.91	25	2.60 ± 0.65	25	1.75 ± 0.44	25	-	-	-	-
6	Biphenyl	1.34 ± 0.58	43	0.80 ± 0.34	43	0.62 ± 0.27	43	0.24 ± 0.10	43	0.42 ± 0.18	43
7	Dibenz[a, h]anthracene	17.28 ± 7.08	41	6.05 ± 2.48	41	10.12 ± 4.15	41	1.80 ± 0.74	41	-	-
8	Pyrene	9.98 ± 2.70	27	6.29 ± 1.70	27	4.86 ± 1.99	41	-	-	0.96 ± 0.39	41
9	Phenanthrene	20.37 ± 5.70	28	1.31 ± 0.54	41	1.31 ± 0.54	41	0.18 ± 0.07	41	-	-
10	Fluorantin	2.56 ± 1.15	45	1.77 ± 0.80	45	1.63 ± 0.73	45	1.22 ± 0.55	45	-	-
11	Fluoren	-	-	-	-	0.51 ± 0.21	42	0.27 ± 0.11	42	0.50 ± 0.21	42
12	Chrysen	2.63 ± 1.18	45	1.57 ± 0.71	45	1.10 ± 0.50	45	0.13 ± 0.06	45	-	-
13	2-methylnaphthalene	-	-	-	-	1.90 ± 0.78	41	1.32 ± 0.54	41	1.67 ± 0.68	41
14	Acenaften	-	-	-	-	-	-	-	-	2.63 ± 1.13	43
15	Benz[g, h, i]perylene	-	-	-	-	-	-	-	-	2.00 ± 0.88	44
16	Naphthalene All	-	-	-	-	-	-	-	-	-	-
	All	72.83		29.17		29.85		6.9		8.32	

**Table 4 life-12-01047-t004:** Viability and RICC of MRC-5 cells that were exposed to PM.

Sample		RICC		Survival Rate, %	
Negative control		100		88.75 ± 3.24	
Dilution control		87.9 ± 6.76		87.80 ± 5.63	
Positive control (1 mg/mL)		26.67 ± 24.55		70.9 ± 6.97	
Sample	Concentration, mg/mL	PM_10_		PM_0.1_	
		RICC	Survival rate, %	RICC	Survival rate, %
C	1	23 ± 6.81	85.33 ± 6.59	30.67 ± 25.66	76 ± 7.35
	0.5	34.17 ± 6.46	83.17 ± 4.99	54.83 ± 24.24	80.17 ± 5.74
	0.25	47.17 ± 27.08	86.00 ± 6.1	48.16 ± 32.42	84.67 ± 6.92
B	1	32.42 ± 21.16	80 ± 9.94	7.41 ± 10.64 *	75 ± 7.12
	0.5	32.17 ± 12.37	84.83 ± 4.07	26.39 ± 13.28	84 ± 6.74
	0.25	34.2 ± 3.27	84 ± 7.48	44.98 ± 38.91	86 ± 3.9
N	1	40.33 ± 3.05	85 ± 5	22.83 ± 2.4	75.17 ± 5.78
	0.5	37 ± 16.09	86.33 ± 1.15	31.28 ± 3.59	84.57 ± 2.99
	0.25	56 ± 9.54	87.33 ± 3.78	39.57 ± 6.5	84.57 ± 2.44
K	1	51.4 ± 12.03 *	84.8 ± 3.49	29.43 ± 9.95 *	79.14 ± 10.11
	0.5	63.4 ± 14.43	89.6 ± 2.88	41.43 ± 8.69	78.71 ± 5.76
	0.25	75 ± 16.67	86.8 ± 4.6	49.71 ± 13.08	81.28 ± 4.07
P	1	42.83 ± 12.16	78.92 ± 3.96	32.83 ± 2.23	75.5 ± 9.65
	0.5	55.73 ± 26.54	84.64 ± 6.39	52.14 ± 19.85	81.28 ± 5.82
	0.25	55.58 ± 32.3	87.17 ± 4.49	50.67 ± 20.59	83.83 ± 3.82

Note: *—Marked differences between high and low concentrations of PM (0.25–1 mg/mL) *p* < 0.01.

**Table 5 life-12-01047-t005:** Results of micronuclei test of MRC-5 cells that were exposed to PM.

Sample	Micronuclei, ‰		Nuclear Protrusion, ‰		Nucleoplasmic Bridge, ‰	
Negative control	27.87 ± 13.73		15.37 ± 9.44		2.87 ± 3.04	
Dilution control	28.8 ± 8.43		15.7 ± 7.41		4.9 ± 4.07	
Positive control (1 mg/mL)	147.4 ± 76.1		113.2 ± 60.71		24.1 ± 12.40	
	PM_10_	PM_0.1_	PM_10_	PM_0.1_	PM_10_	PM_0.1_
C	26.05 ± 14.09	65.17 ± 31.49	20.11 ± 15.01	42.87 ± 19.22	4.83 ± 4.02	13.39 ± 10.72
B	48.29 ± 31.55	96.52 ± 40.06	34.97 ± 31.1	58.18 ± 46.77	10.47 ± 9.24	20.89 ± 11.41
N	71.17 ± 28.99	68.72 ± 43.74	78.89 ± 21.17	71.57 ± 43.09	17.89 ± 9.65	21.82 ± 8.87
K	48.77 ± 22.76	53.83 ± 35.4	51.60 ± 25.4	67.93 ± 36.72	17.15 ± 9.96	22.51 ± 17.15
P	39.34 ± 24.08	76.13 ± 24.81	44.34 ± 24.37	62.34 ± 17.56	12.94 ± 8.11	22.58 ± 15.81

## Data Availability

Not applicable.

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
