# Peer review of "Chemical Composition and Toxicity of PM10 and PM0.1 Samples near Open-Pit Mines and Coal Power Stations"

_life, 2022, doi:10.3390/life12071047_

Round 1
Reviewer 1 Report
The reviewed manuscript deals with the topic of fine and ultrafine particulate matter originating from coal industry and its influence on the human respiratory tract in an in-vitro experiment. This topic seems to stay relevant as long as people continue to mine and burn coal.
General comments:
- The methods used and the results obtained seem adequate, but the discussion is lacking. Many areas of the manuscript need improvement.
- The Conclusions section is missing.
- Graphical abstract is missing.
- English language: The manuscript requires proof-reading. It is generally well-written, but the quality is uneven between sections. Showing the text to a native English speaker would certainly increase its quality. There are sentences with many mistakes especially in the Introduction which leaves a bad taste.
- All acronyms should be defined in the text.
Abstract
- It seems that there is a contradiction in the distance from the polluting industries between the Abstract, Introduction (1-1.5 km) and the Methods (2.5-3 km).
- “The study findings suggest that UFPs deserve special attention as a biological agent, distinct from larger PMs” – this is an important finding which needs to be discussed in greater detail in the Discussion and of course in the Conclusions.
Introduction
- The Introduction would benefit from a better structure. Please consider proof-reading and revising it. Right now it leaves the following impression: the authors start with the aerosol emissions from mining, then add the landscape factors. Then without explaining the difference between aerodynamic sizes (diameters) of PM, the authors suddenly switch to discussing nanoparticles (lines 48-54). Then the story goes to characteristics of coal and finally explains the difference in sizes of PM, etc.
- It is also necessary to show here if similar in-vitro experiments with UFP and lung cells were conducted before, and how this study could inmprove the existing knowledge.
- Lines 41-42. Rephrase? The meaning is not clear. No safe PM exposure limits exist?
- Lines 43-47. Unrelated factors are lumped together in one paragraph. What do landscape, humidity, and drilling have in common? Also, “input of PM” – input to what? Overall air pollution?
- Lines 48-54. The paragraph about nanoparticles seems out of place here. It would be better to first define the size classes of PM. Define fine respirable PM, UFPM, etc. Also, the paragraph itself should be revised.
- Lines 57-58. Does it have anything to do with the current study?
- Lines 64-65. Is using the word “rocks” OK in this context?
- Lines 67-71. This fragment should be up higher.
- Lines 80-83. Without any comparison to the data from other regions this is hard to follow. Is it good of bad? The referenced study says their figures are considerably higher than other studies, but it is not clear from your paragraph. Please consider revising this statement.
- Lines 84-91. This paragraph seems to be very location-dependant. In the region where this reviewer lives, the wind speed in winter can be very high. Is this information only specific for India? How does it relate to coal/weather situation in Kemerovo? Please consider revising this paragraph.
- Line 85. “Greatly reduction” – just a single example of phrases that need to be cleaned up in the manuscript.
- Line 108 and elsewhere. “Coal mining object” – try to find a better synonym. Maybe, facility?
- Line 108. “…capable of exhibiting…” – I’m not sure “exhibiting” is the right word here.
- Lines 109-110. Why past tense all of a sudden? Found by who?
- Line 114. Why exactly this distance was chosen? UFPs can travel extremely large distances.
Materials and Methods
- Sampling, preparation and analysis methods seem adequate. The results of Raman spectroscopy are presented in the Results section, but the description of this method is not given in the Materials and Methods.
- A general map of the area with the location of the said industrial facilities could be included.
- At least a brief description of these facilities should be given to the readers, showing the official data on how much coal they produce/burn, how they influence the environment of nearby populated areas, etc.
- Line 125. In the Abstract and Introduction it says 1-1.5 km from the border of open cast mine. Which one is right?
- Lines 142-144. What is the reason behind freezing samples and then immediately thawing them?
Results
- This section would really benefit from having illustrative material. Surely there were some interesting visual findings during the experiments. Some microscopic shots of stand-out elements, like for example in this study: https://doi.org/10.3390/ijerph18179234; or comparison of Raman spectra like here: https://doi.org/10.1016/j.heliyon.2020.e03299; or maybe charts showing growth inhibition and/or mortality of cells like in this study: https://doi.org/10.1016/j.toxrep.2021.04.004.
Discussion
- The Discussion has several paragraphs of general information which are more suited for the Introduction or a literature review. If the authors wish to include this data, it should be a) relevant for this section; b) debated or compared with the obtained data.
- The importance of the findings of the authors should be discussed and stressed in this section.
- Lines 391-400. This paragraph seems to be more suited for the Introduction.
- Lines 409-410. Not a correct thing to say because solid particles are always there.
- Lines 422-428. This a very general paragraph. What does in add to the discussion?
- Lines 44-451. Another very general statement.
- Line 454. What is RM?
- Lines 462-464. Since these are not debated or compared with, aren't they better suited for the Introduction?
- Lines 467-480. Does this study deal with the DNA? Are these two paragraphs relevant?
- The Discussion seems unfinished.
Author Response
The reviewed manuscript deals with the topic of fine and ultrafine particulate matter originating from coal industry and its influence on the human respiratory tract in an in-vitro experiment. This topic seems to stay relevant as long as people continue to mine and burn coal.
General comments:
- The methods used and the results obtained seem adequate, but the discussion is lacking. Many areas of the manuscript need improvement.
- The Conclusions section is missing.
A: We add the Conclusions section
- Graphical abstract is missing.
A: We add the graphical abstract
- English language: The manuscript requires proof-reading. It is generally well-written, but the quality is uneven between sections. Showing the text to a native English speaker would certainly increase its quality. There are sentences with many mistakes especially in the Introduction which leaves a bad taste.
A: Manuscript was reproof-readed, particulary Introduction section.
- All acronyms should be defined in the text.
A: Corrected
Abstract
- It seems that there is a contradiction in the distance from the polluting industries between the Abstract, Introduction (1-1.5 km) and the Methods (2.5-3 km).
A: We replaced this point at Materials and methods section at Line 125: 1-2,5 km is right. There and further in all text remove/replace operation we used lines number taken from original manuscript.
- “The study findings suggest that UFPs deserve special attention as a biological agent, distinct from larger PMs” – this is an important finding which needs to be discussed in greater detail in the Discussion and of course in the Conclusions.
A: We add the Conclusions section
Introduction
- The Introduction would benefit from a better structure. Please consider proof-reading and revising it. Right now it leaves the following impression: the authors start with the aerosol emissions from mining, then add the landscape factors. Then without explaining the difference between aerodynamic sizes (diameters) of PM, the authors suddenly switch to discussing nanoparticles (lines 48-54). Then the story goes to characteristics of coal and finally explains the difference in sizes of PM, etc.
A: We rearranged Introduction section. Mining specificity points now prior to the text for particles-related paragraphs. Lines 48-54 paragraph we move after 106 line of original manuscript.
Also lines 84-91 paragraph moved after line 54 of original manuscript.
- It is also necessary to show here if similar in-vitro experiments with UFP and lung cells were conducted before, and how this study could inmprove the existing knowledge.
A: We add an additional paragraph after line 106 of the original manuscript to show UFP in vitro experiments. Also some pro/contra of experiment design using snow-sampled PM considered in Discussion section.
UFP exposure often discussed in connection to cells inflammatory response and oxidative stress induction, pathological response of lung tissue can include epithelial-mesenchymal transition [Cochard, 2020]. UFP exposure can change gene expression. Diesel and biomass combustision UFP can induce expression of inflammatory markers and transcriptional markers relevant for cardiovascular disease [Grilli, 2018] alteration in mitochondric metabolism and induction of cytochromic genes [Borgie, 2015] in human BEAS-2b cells. Ultra fine carbonaceous particles can induce heme oxygenase-1 but not IL-6 and IL-8 [Bitterle, 2006], diesel exhaust UFP can increase metal oxide cytotoxicity [Zerboni, 2019] in A549 human lung adenocarcinoma cells. Also reactive oxygene species generation following UFP exposure demonstrated in mouse pulmonary microvascular endothelial cells [Mo, 2009] and protein oxidation and DNA adduct in mouse in vivo [Aztatzi, 2018].
- Lines 41-42. Rephrase? The meaning is not clear. No safe PM exposure limits exist?
A: We removed this contradictory thesis.
- Lines 43-47. Unrelated factors are lumped together in one paragraph. What do landscape, humidity, and drilling have in common? Also, “input of PM” – input to what? Overall air pollution?
A: Lines 44-47 sentence was changed to:
The features of the surrounding natural and anthropogenic landscape can affect the amount of PM generated at any point.
Following phrase: “which can sharply increase input of PM compared with the calculated value” was removed.
- Lines 48-54. The paragraph about nanoparticles seems out of place here. It would be better to first define the size classes of PM. Define fine respirable PM, UFPM, etc. Also, the paragraph itself should be revised.
A: We rearranged Introduction section as it mentioned above.
- Lines 57-58. Does it have anything to do with the current study?
A: Following sentence was removed: However, it is complicated to repeat the conditions of rock crushing that occurs under natural conditions in an experimental set-up.
- Lines 64-65. Is using the word “rocks” OK in this context?
A: “of such rocks” phrase was removed.
- Lines 67-71. This fragment should be up higher.
A: 62-67(Coal-bearing rocks are characterized by a high level of erosion and the ability to form respirable PM10 and PM2.5. And a greater extent, this applies to lignite, which increases dust formation several times due to wind erosion as well as during transportation [6]. Moreover, incomplete combustion of coal leads to generation of coarse PM10, which includes coal particles and inorganic calcium-alumo-silicate fragments.) added to 55-61 paragraph. (In regions that lie at close proximity to coal mines the geological features and properties of coal can considerably affect the amount and quality of generated dust [5]. The main characteristics of the coal and surrounding rocks are their brittleness and hardness. An increase in brittleness leads to an increase in the proportion of fine and respirable PM and decreases the mass fraction of dust relative to the mass of coal and rock [5]. Coal-bearing rocks are characterized by a high level of erosion and the ability to form respirable PM10 and PM2.5. And a greater extent, this applies to lignite, which increases dust formation several times due to wind erosion as well as during transportation [6]. Moreover, incomplete combustion of coal leads to generation of coarse PM10, which includes coal particles and inorganic calcium-alumo-silicate fragments.)
67-71 sentences was moved to 2nd Introduction paragraph after line 47 in original manuscript.
- Lines 80-83. Without any comparison to the data from other regions this is hard to follow. Is it good of bad? The referenced study says their figures are considerably higher than other studies, but it is not clear from your paragraph. Please consider revising this statement.
A: We add next sentences after line 83.
“The authors acknowledge the obtained values as high in comparison with a number of similar studies. The generation of PM at the studied sites can be quite high, given the high intensity of production (5-9 million tons per year at each open-pit mine).”
- Lines 84-91. This paragraph seems to be very location-dependant. In the region where this reviewer lives, the wind speed in winter can be very high. Is this information only specific for India? How does it relate to coal/weather situation in Kemerovo? Please consider revising this paragraph.
A: Sentences 1-4 of this paragraph depends to the region of study.
We add to the line 84 of original manuscript. “In the study area”
- Line 85. “Greatly reduction” – just a single example of phrases that need to be cleaned up in the manuscript.
A: We agreed, comparative degree “a great” was removed.
- Line 108 and elsewhere. “Coal mining object” – try to find a better synonym. Maybe, facility?
A: Replaced to “coal mine”
- Line 108. “…capable of exhibiting…” – I’m not sure “exhibiting” is the right word here.
A: Replaced to exposing
- Lines 109-110. Why past tense all of a sudden? Found by who?
A: Replaced to “Metals, polycyclic aromatic hydrocarbons (PAHs), and quartz particles also can be found in different proportions in PM samples [Ishtiaq, 2018].”
- Line 114. Why exactly this distance was chosen? UFPs can travel extremely large distances.
Materials and Methods
- Sampling, preparation and analysis methods seem adequate. The results of Raman spectroscopy are presented in the Results section, but the description of this method is not given in the Materials and Methods.
A: We add Raman spectroscopy description to materials and methods section.
- A general map of the area with the location of the said industrial facilities could be included.
A:We add location of sample points to Material and methods section.
- At least a brief description of these facilities should be given to the readers, showing the official data on how much coal they produce/burn, how they influence the environment of nearby populated areas, etc.
A: We add description to 2.1 section.
- Line 125. In the Abstract and Introduction it says 1-1.5 km from the border of open cast mine. Which one is right?
A: We made an appropriate correction
- Lines 142-144. What is the reason behind freezing samples and then immediately thawing them?
A: After mixing each snow sample stored in freeze from 1 to 2 weeks, because filtration took some time. To correct procedure description in line 144 we delete “immediately”.
Results
- This section would really benefit from having illustrative material. Surely there were some interesting visual findings during the experiments. Some microscopic shots of stand-out elements, like for example in this study: https://doi.org/10.3390/ijerph18179234; or comparison of Raman spectra like here: https://doi.org/10.1016/j.heliyon.2020.e03299; or maybe charts showing growth inhibition and/or mortality of cells like in this study: https://doi.org/10.1016/j.toxrep.2021.04.004.
Discussion
- The Discussion has several paragraphs of general information which are more suited for the Introduction or a literature review. If the authors wish to include this data, it should be a) relevant for this section; b) debated or compared with the obtained data.
- The importance of the findings of the authors should be discussed and stressed in this section.
- Lines 391-400. This paragraph seems to be more suited for the Introduction.
A: This paragraph moved to introduction section.
- Lines 409-410. Not a correct thing to say because solid particles are always there.
A: We removed this contradictory sentence.
- Lines 422-428. This a very general paragraph. What does in add to the discussion?
A: We add this paragraph as important part of discussion concerned to PAH found in thawed water samples, but not in PM itself. PAH level differences, showed in present research can reflects storage conditions and other features in open-pit mines.
- Lines 444-451. Another very general statement.
A: We found sufficient to mark the fact that not only mines itself but vehicle traffic contribute to PM emission, that greatly increase an area of potential residents exposure. Residential proximity to traffic ways is very actual for research region.
- Line 454. What is RM?
A: Corrected to PM.
- Lines 462-464. Since these are not debated or compared with, aren't they better suited for the Introduction?
A: We move this paragraph to Introduction section after paragraph concerned with UFP in vitro cells adverse effects.
- Lines 467-480. Does this study deal with the DNA? Are these two paragraphs relevant?
A: Biology response assessed by different genotoxicity tests, based on primary DNA damage. As micronucleus assay based on counts chromosome-based DNA-originated fragments, so in DNA comet assay measures fragmentation level. We suggest all discussed research used indicators reflects DNA direct/indirect damage and relevant for our results.
The Discussion seems unfinished.
A: We have moved sentence at line 481-483 upper after line 461 of original manuscript.

Reviewer 2 Report
The authors present a study on the chemical composition and cell toxicity of PM collected from samples near open cut mines and power station in the Kemerovo region of Russia. The authors should provide a map of the study area with location of open cut mines and power stations as well as the sample points as many readers are not familiar with the area. The authors should also explicitly state the aim of the study in the introduction. There is no conclusion section in the manuscript which state whether the study achieve its aim and the implication of the results.
I recommend the manuscript to be accepted for publication after a revision to address the above points and the following minor comments
(1) All the size xx in PMxx specified in the manuscript should be written as subscripted
(2) Line 55: comma after "coal mines" i.e "coal mines, "
(3) Line 75: "(50–100 m," should be "(50–100) m, "
(4) Line 83: "6*10-9 - 4*10-8" should have index as superscript. What is the unit of these values ?
(5) Line 86: Better words for "lower air layers" are "lower atmospheric boundary layers"
(6) Line 91: "PM 1.0". PM1.0 or PM10 ? and number should be subscripted
(7) Line 122: should be sub-heading "Collection and extraction of PM" as "2.1 Collection and extraction of PM"
(8) Line 126: rather than "ray", "direction" should be used. The wind rose should be provided in the manuscript
(9) Line 170: "Physical and chemical analysis of the melting snow and particulate matter samples.". This should be a sub-heading with proper order number i.e "2.2 Physical and chemical analysis of the melting snow and particulate matter samples"
(10) Line 172: "Absorption spectroscopy of the melted snow samples". Incomplete sentence
(11) Line 198, 204, 253, 274: Again sub-heading 2.3 , 2.4, 2.5, 2.6
(12) Line 382: "from" not "form"
(13) Line 391: "PM <10 μm," should be "PM less than 10 μm,"
(14) Line 483: Should have a conclusion section specifying the aim of the study, whether the study has achieved the aim and recommendation.
Author Response
The authors present a study on the chemical composition and cell toxicity of PM collected from samples near open cut mines and power station in the Kemerovo region of Russia.
The authors should provide a map of the study area with location of open cut mines and power stations as well as the sample points as many readers are not familiar with the area.
The authors should also explicitly state the aim of the study in the introduction.
There is no conclusion section in the manuscript which state whether the study achieve its aim and the implication of the results.
I recommend the manuscript to be accepted for publication after a revision to address the above points and the following minor comments
(1) All the size xx in PMxx specified in the manuscript should be written as subscripted
A: Corrected
(2) Line 55: comma after "coal mines" i.e "coal mines, "
A: Corrected
(3) Line 75: "(50–100 m," should be "(50–100) m, "
A: Corrected
(4) Line 83: "6*10-9 - 4*10-8" should have index as superscript. What is the unit of these values ?
A: We cited this as dimensionless coefficient as a share of PM of aerodynamically size to total coal/rocks processed in mine. For example 1*10-9 corresponds 1 milligram PM per 1 ton.
(5) Line 86: Better words for "lower air layers" are "lower atmospheric boundary layers"
A: Corrected
(6) Line 91: "PM 1.0". PM1.0 or PM10 ? and number should be subscripted
A: Corrected
(7) Line 122: should be sub-heading "Collection and extraction of PM" as "2.1 Collection and extraction of PM"
A: Corrected
(8) Line 126: rather than "ray", "direction" should be used. The wind rose should be provided in the manuscript
A: Corrected. Wind rose added to Figure 1 in 2.1 section.
(9) Line 170: "Physical and chemical analysis of the melting snow and particulate matter samples.". This should be a sub-heading with proper order number i.e "2.2 Physical and chemical analysis of the melting snow and particulate matter samples"
A: Corrected
(10) Line 172: "Absorption spectroscopy of the melted snow samples". Incomplete sentence
A: Corrected
1st and 2nd sentences were corrected to: “Absorption spectroscopy of the melted snow samples that were concentrated eight times were registered using a scanning spectrophotometer Shimadzu UV-3600 (Shimadzu Corporation, Kyoto, Japan).”
(11) Line 198, 204, 253, 274: Again sub-heading 2.3 , 2.4, 2.5, 2.6
A: Corrected
(12) Line 382: "from" not "form"
A: Corrected
(13) Line 391: "PM <10 μm," should be "PM less than 10 μm,"
A: Corrected
(14) Line 483: Should have a conclusion section specifying the aim of the study, whether the study has achieved the aim and recommendation.
A: Corrected
We add a Conclusion section.
